materials science/nanotechnology/computational chemistry

zeolite, high pressure, X-ray diffraction, symmetry, phase transition, cubic

**Author for correspondence:**
Asel Sartbaeva
e-mail: a.sartbaeva@bath.ac.uk

This article has been edited by the Royal Society of Chemistry, including the commissioning, peer review process and editorial aspects up to the point of acceptance.

# Pressure-induced symmetry changes in body-centred cubic zeolites

Antony Nearchou[1], Mero-Lee U. Cornelius[3],
Zöe L. Jones[1], I. E. Collings[4], Stephen A. Wells[2],
Paul R. Raithby[1] and Asel Sartbaeva[1]

[1]Department of Chemistry, and [2]Department of Chemical Engineering, University of Bath, Claverton Down, Bath BA2 7AY, UK
[3]Department of Chemistry, University of the Western Cape, Bellville, Cape Town 7535, South Africa
[4]European Synchrotron Radiation Facility, 71 avenue des Martyrs, 38000 Grenoble, France

SAW, 0000-0002-3920-3644; PRR, 0000-0002-2944-0662;
AS, 0000-0003-1017-0161

Previous work has shown a strong correlation between zeolite framework flexibility and the nature of structural symmetry and phase transitions. However, there is little experimental data regarding this relationship, in addition to how flexibility can be connected to the synthesis of these open-framework materials. This is of interest for the synthesis of novel zeolites, which require organic additives to permutate the resulting geometry and symmetry of the framework. Here, we have used high-pressure powder X-ray diffraction to study the three zeolites: Na-X, RHO and ZK-5, which can all be prepared using 18-crown-6 ether as an organic additive. We observe significant differences in how the occluded 18-crown-6 ether influences the framework flexibility—this being dependent on the geometry of the framework. We use these differences as an indicator to define the role of 18-crown-6 ether during zeolite crystallization. Furthermore, in conjunction with previous work, we predict that pressure-induced symmetry transitions are intrinsic to body-centred cubic zeolites. The high symmetry yields fewer degrees of freedom, meaning it is energetically favourable to lower the symmetry to facilitate further compression.

## 1. Introduction

Zeolites are a class of microporous aluminosilicates, recognized for possessing periodic open-framework structures. On the molecular level, zeolites are built from corner-sharing tetrahedra of the form $TO_4$, where the T atoms are Si or Al. From a topological perspective, these primary tetrahedra arrange into regular

**Figure 1.** The structures and space groups of zeolites Na-X, ZK-5 and the C- and A-forms of zeolite RHO. The framework topology is indicated in parentheses. Oxygen atoms are shown as red spheres, and SiO$_4$ tetrahedra as translucent solids.

geometric cages and rings, which are referred to as secondary building units (SBUs) [1,2]. Due to the presence of tetracoordinated aluminium, the framework retains a net negative charge, which is counter-balanced by mobile metal cations dispersed throughout the framework. Conventional solid-state chemistry considers solids as static materials; however, zeolite frameworks demonstrate an inherent flexibility [3–7]. Although the TO$_4$ tetrahedral units are rigid, the T-O-T bridging angles possess significant freedom, permitting the framework to contract or expand as a response to thermodynamic stimuli [1,8]. Such low-frequency dynamics can result in distortions of the underlying SBUs and in some cases alterations in symmetry.

Little research is reported on the low-frequency modes and elastic behaviour of zeolites under compression. This is unfortunate, as such research is shown to be beneficial in understanding how atomic-scale topological structure can dictate physical properties. A prime instance of this is expressed by the family of fibrous zeolites, specifically edingtonite, natrolite, thomsonite and scolecite [9,10]. All four zeolites display anisotropic elastic behaviour, which is dictated by the anti-rotation of the SBU chains present in their structures. The nature of how geometry and SBUs influence elastic behaviour has been explored through the concept of the flexibility window [3]. This being the range of densities at which the framework can exist where the connectivity is retained and the TO$_4$ tetrahedral units remain rigid and undeformed. The use of the flexibility window has proven to explain pressure-induced phase transitions [11–13], in addition to being a potential criterion to rationally design novel zeolites [5].

Furthermore, high-pressure studies provide scope for alternative preparation routes to novel zeolites, via reconstructive phase transitions [14]. However, several zeolites express pressure-induced phase transitions that do not require disassembly of the framework. Such transitions appear to be more intrinsic to the geometry of the framework. Such an example is zeolite RHO, which possesses an ambient centric space group of *Im-3m* that evolves into the acentric *I-43m* space group with the application of pressure [15–17]. This is characterized by the increasing ellipticity of the eight-ring openings in the framework, as shown in figure 1.

**Table 1.** Hydrogel batch compositions used in the preparation of zeolites Na-X, RHO and ZK-5.

| zeolite | $Al_2O_3$ | $Na_2O$ | $K_2O$ | $Cs_2O$ | SrO | $SiO_2$ | 18C6 | $H_2O$ |
|---------|-----------|---------|--------|---------|-----|---------|------|--------|
| Na-X | 1 | 2.9 | | | | 10 | 0.5 | 90 |
| RHO | 1 | 1.8 | | 0.3 | | 10 | 0.5 | 100 |
| ZK-5 | 1 | | 2.7 | | 0.1 | 10 | 1.0 | 220 |

This phase transition has also been observed with specific dehydration and heating conditions [18,19]. Another example concerns ANA-type zeolites, of which the materials analcime [20], leucite [21], pollucite [22] and wairakite [23] all show pressure-induced reductions in symmetry to a triclinic system. Such phase transitions have shown good agreement with the flexibility window property [12,24], whereby the transitions are defined and facilitated by the nature of the topological geometry. Interestingly, for both RHO and ANA-type zeolites, the data consistently demonstrate that the higher pressure polymorph is more compressible [16,20–23].

The mechanical behaviour of zeolites is also strongly influenced by the presence of extra-framework content and the pressure-transmitting medium used. These media are characterized as *penetrating* and *non-penetrating* fluids [10], whereby the use of penetrating fluid typically increases the mechanical stiffness due to increased extra-framework content [25–28]. These observations are a result of the pressure-transmitting medium occupying the pores of the framework structure and essentially preventing structural collapse. This is evident from both experimental behaviour [11] and the simulated flexibility window of zeolites [29]. The importance of considering such framework content is emphasized by the over-hydration effect, whereby non-framework content such as water is being forced into the framework with the application of pressure [30–32]. Consequently, this leads to a decrease in compressibility as a function of pressure. However, other work has shown how the intrinsic flexibility window and compression of the EMT framework is unperturbed by the occlusion of 18-crown-6 ether (18C6) [33,34]. This is of particular interest, as 18C6 is used as an organic additive in the preparation of EMT-type zeolites like EMC-2 [35,36], where synthesis without 18C6 has not been achieved [37]. Our earlier high-pressure and computational studies suggest that 18C6 may not necessarily behave as a true geometric template but rather influences the free-energy landscape of crystallization [34]. This highlights how the study of framework flexibility can provide valuable insights into the nature of zeolite synthesis.

Figure 1 shows the framework structure of the cubic zeolites Na-X, RHO and ZK-5. Like zeolite EMC-2, all three can be prepared using 18C6 as an organic additive [36]. In zeolite Na-X, the 18C6 molecule occupies the *t-fau* supercage, whereas in RHO and ZK-5, it occupies the α-cage. Both zeolites Na-X and RHO have been investigated under high pressure [15–17,26,38], with the latter displaying the aforementioned phase transition. However, zeolite ZK-5, which is topologically analogous to zeolite RHO, has not been studied previously as far as we know. In addition to this, the influence of the occluded 18C6 on the framework dynamics has likewise not been studied for all three zeolites. Herein, we report for the first time experimental high-pressure data for zeolite ZK-5, in addition to the as-synthesized zeolites Na-X, RHO and ZK-5 with 18C6 occluded in the framework cavities. From these data, we have gleaned insights into the role of 18C6 in the crystallization of these three zeolites. Furthermore, we propose that pressure-induced changes in symmetry are an intrinsic feature to body-centred cubic zeolites. Such reductions in symmetry are favourable as they facilitate further compression without deformation of the framework tetrahedra, preventing 'early onset' pressure-induced amorphization.

# 2. Material and methods

## 2.1. Sample preparation

The zeolite Na-X, RHO and ZK-5 samples used in the high-pressure analysis were synthesized following the procedures used by Chatelain *et al.* [36,39,40]. The molar batch compositions of the precursor hydrogel for each synthesis are shown in table 1. The materials used were sodium hydroxide (NaOH), potassium hydroxide (KOH), caesium hydroxide solution (50 wt% CsOH in water), strontium nitrate ($Sr(NO_3)_2$), 18-crown-6 ether ($C_{12}H_{24}O_6$ 18C6), sodium aluminate ($NaAlO_2$), aluminium hydroxide ($Al(OH)_3$), colloidal silica (LUDOX® HS-40, 40 wt% $SiO_2$ in water) and distilled water. All materials were purchased from Sigma-Aldrich.

### 2.1.1. Zeolite Na-X

In a Naglene Teflon FEP bottle, the sodium hydroxide and 18C6 were dissolved in the distilled water. The sodium aluminate was subsequently added to the solution and stirred until homogeneous. Next, the colloidal silica was slowly added to the solution while stirring to avoid rapid gelation. The hydrogel was then aged for 4 h under ambient conditions, before being sealed and the bottle placed into a 100°C oven for 8 days. Upon the completion of crystallization, the bottle was removed from the oven, cooled and the product separated from the mother liquor using Buchner filtration. The product was washed with distilled water, until the filtrate was of neutral pH. The product powder was subsequently dried and ground until sample calcination and dehydration.

### 2.1.2. Zeolite RHO

The sodium hydroxide and 18C6 were dissolved in the distilled water and caesium hydroxide solution. Upon dissolution, the sodium aluminate was added and left to stir until the solution was homogeneous. Following this, the colloidal silica was poured into the solution slowly so as not to produce a viscous gel. The formed hydrogel was then aged for 24 h under stirring at ambient conditions. After ageing, the gel was transferred to a Teflon cup within a sealed stainless-steel autoclave. The hydrogel-containing autoclave was subsequently placed into a 110°C oven for 8 days. After this time, the autoclave was removed from the oven, cooled and opened. The product was removed from the mother liquor using Buchner filtration and washed with distilled water until the filtrate was of neutral pH. The washed powder was then dried and ground until sample calcination and dehydration.

### 2.1.3. Zeolite ZK-5

Within a conical flask, the potassium hydroxide was dissolved in distilled water. The aluminium hydroxide was added to this solution and the flask weighed. The solution was then heated to near 110°C under stirring. Upon dissolution of the aluminium hydroxide, the solution was cooled, weighed and any water lost during heating topped up with additional water. In a separate beaker, the strontium nitrate and 18C6 were dissolved in a small amount of water, followed by the colloidal silica, ensuring the solution was homogeneous. Subsequently, the alumina solution was quickly added to the silica solution under stirring, to form the hydrogel. The hydrogel was stirred for 30 min to ensure it was thoroughly mixed.

The gel was then transferred to a Teflon cup within a sealed stainless-steel autoclave. Next, the autoclave was placed into a 150°C oven for 5 days. Upon complete crystallization at this time, the autoclave was removed from the oven and allowed to cool. The crystalline product was then separated from the mother liquor via Buchner filtration and washed until the filtrate was of neutral pH. The product was then dried and suitably ground until sample calcination and dehydration.

### 2.1.4. Calcination and dehydration

The filled (18C6 containing) and empty (calcined) analogues of each zeolite were prepared from the same sample that was separated into two portions.

To prepare the empty analogue of each zeolite, half of the sample was calcined in air. The powder was heated at a ramp rate of 1 K min$^{-1}$ to 100°C, 200°C and 300°C for 1 h and finally 450°C for 6 h. After calcination, the sample was cooled at a rate of 1 K min$^{-1}$, stopping at 200°C for 1 h and terminating at ambient temperature.

The calcined and as-synthesized analogue of each zeolite was next dehydrated under vacuum. The samples were heated at a ramp rate of 1 K min$^{-1}$, holding at 100°C for 1 h and 200°C for 6 h. After dehydrating at 200°C, the sample was cooled at a rate of 1 K min$^{-1}$, stopping at 100°C for 1 h and terminating at ambient temperature.

## 2.2. High-pressure X-ray diffraction

The zeolite powder samples were analysed using high-pressure X-ray diffraction on the ID15B beamline at the European Synchrotron Radiation Facility (ESRF) in Grenoble, France. The samples were loaded into diamond anvil cells (DACs) alongside a ruby chip, using Daphne 7373 oil as a non-penetrating pressure-transmitting medium. The applied pressure in the DAC was determined by the shift of the R1 emission line of the ruby fluorescence [41]. The incident synchrotron X-ray radiation used in

the experiment was of wavelength 0.4113 Å, and the detector parameters were calibrated using silicon. At sequential steps of increasing pressure, a diffraction pattern was taken, with the pressure being recorded before and after each pressure step to provide an average pressure throughout the measurement. The samples were compressed until complete pressure-induced amorphization was imminent. The DACs were then depressurized, with several diffraction patterns measured upon the decompression cycle to ambient conditions. The experimental error in pressure was estimated at 0.1 GPa in the 0.0–3.0 GPa region of the compression cycle. Above 3.0 GPa and throughout the entirety of the decompression cycle, the error was estimated at 0.4 GPa.

Three diffraction images were taken at each pressure point, which were used to produce an average image using the FIT2D software [42]. Subsequently, the Dioptas software was used to integrate the two-dimensional area images into one-dimensional powder diffraction patterns [43]. To determine the unit cell parameters at each pressure, Pawley refinements of the diffraction patterns were achieved using the TOPAS Academic software [44]. The sequential refinements with increasing pressure were performed using the Batch mode. Zeolite Na-X was refined to the *Fd-3m* space group. The C- and A-forms observed in zeolite RHO were refined to the *Im-3m* and *I-43m* space groups, respectively. The cubic and tetragonal phases of zeolite ZK-5 were refined to space groups *Im-3m* and *I4/mmm*, accordingly. The error in the calculated unit cell parameters was determined in the TOPAS Academic software. It was seen that the errors were consistently less than 0.004 Å and had a negligible influence on the trends observed. Tables containing the full list of calculated errors for the unit cell parameters and cell volumes are contained in the electronic supplementary material.

The flexibility windows of the empty and filled zeolite frameworks were determined using the GASP software developed by Wells & Sartbaeva [45,46]. The bulk moduli of the zeolite samples were calculated in the PASCal webtool by Cliffe & Goodwin [47]. Only unit cell data within the 0–2.2 GPa range were used and fitted to the second- and third-order Birch–Murnaghan equations of state, depending on which equation showed the best fit. The fits were weighted with the 0.1 GPa error in pressure determination.

# 3. Results

## 3.1. Zeolite Na-X

The unit cell parameter $a$ as a function of pressure for the calcined (empty) and 18C6 containing (filled) zeolite Na-X are shown in figure 2*a,b* accordingly. In both samples, the cell parameter contracts linearly with pressure but displays a small decrease in gradient after approximately 2.2 GPa. This change in gradient is probably a consequence of the Daphne 7373 oil solidifying [48]. The linear unit cell compression is comparable to the results seen by Colligan *et al.* [26] for other FAU-type zeolites of varying Si/Al ratio in different pressure-transmitting media.

The empty zeolite Na-X was compressed to 3.8 GPa, where the onset of pressure-induced amorphization was imminent. Upon decompression, we observed that the unit cell expanded following the same pathway observed with contraction. Throughout the entirety of the experiment, the empty zeolite remained within the flexibility window, indicating that there were no deformations of the rigid tetrahedra. Furthermore, this demonstrates the intrinsic reversible mobility of the framework with compression and decompression within the studied pressure range.

As for the filled zeolite Na-X, the simulated flexibility window is substantially narrowed. This demonstrates that the steric bulk of the 18C6 molecule theoretically restricts the extent to which the framework tetrahedra can move. The experimental data illustrates that the zeolite leaves the confines of the flexibility window, suggesting that the framework tetrahedra are distorted above approximately 2.8 GPa. The filled sample was compressed to 5.1 GPa, and with subsequent decompression, we observed a hysteresis in the expansion of the unit cell. Such a delay in expansion during the decompression cycle can be due to two possibilities. The first is that the final pressure experienced by the filled zeolite was higher than that of the empty zeolite (5 GPa versus 3.8 GPa), which could lead to greater non-hydrostatic stress experienced for the filled zeolite, leading to larger hysteresis upon decompression. The second possibility concerns the potentially distorted framework tetrahedra. During the decompression, these tetrahedra take longer to reassert their ideal tetrahedral geometry, alongside the crystal periodicity. This observation agrees with work by Huang [49] and Havenga *et al.* [50] which report that FAU-type zeolites can express 'structural memory', whereby the extra-framework content permits reversible amorphization.

Figure 3 displays the comparison in unit cell volume contraction with pressure for the empty and filled zeolite Na-X samples. It can be seen that the filled zeolite Na-X sample was capable of being

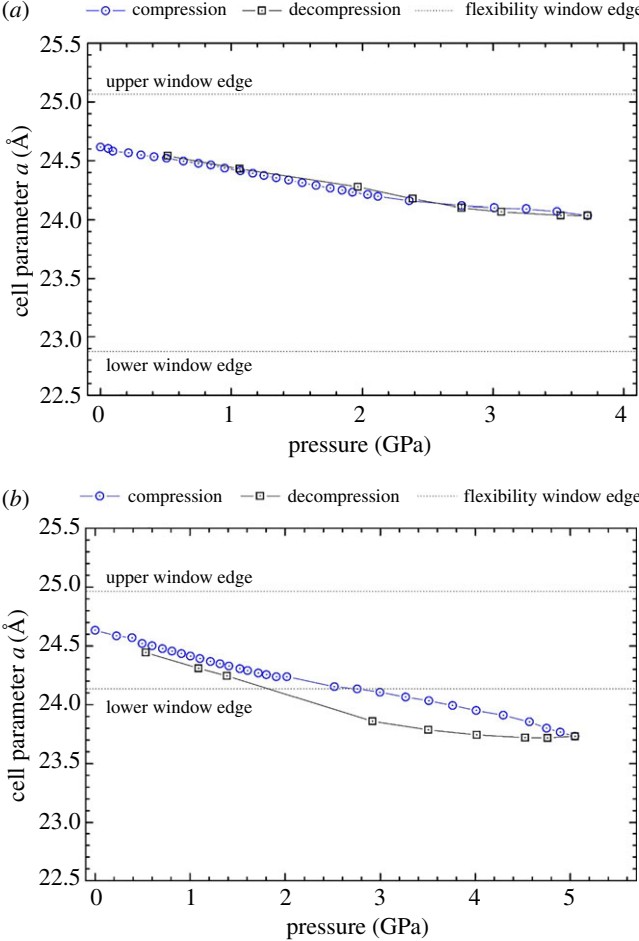

**Figure 2.** Cell parameter *a* as a function of pressure for (*a*) the empty and (*b*) the filled zeolite Na-X. Blue circle data points correspond to the compression cycle, and black squares to the decompression. Also shown are the edges of the flexibility window as simulated in the GASP software.

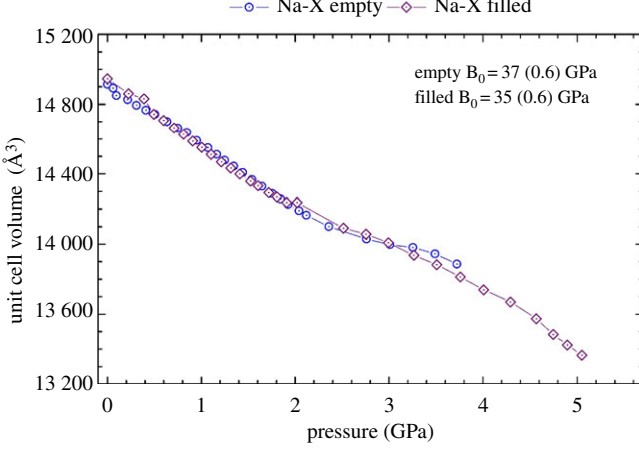

**Figure 3.** The variation of cell volume with increasing pressure for zeolite Na-X. The empty analogue is shown with blue circles, and the filled with purple diamonds. Also included are the bulk moduli ($B_0$) of the two zeolites, using the Birch–Murnaghan second-order fit calculated using the PASCal webtool, as shown in the electronic supplementary material.

compressed to a higher pressure, showing a slower decline of crystallinity with pressure. This indicates that the 18C6 molecule is enhancing the structural integrity of the FAU framework, improving the resistance to pressure-induced amorphization. Such a trend has been reported previously with extra-framework content in the channels of other zeolites [11,13].

Also shown in figure 3 are the calculated bulk moduli of 37 and 35 GPa for the empty and filled zeolite Na-X samples, respectively. These values demonstrate that the mechanical compressibility of the FAU framework is not impeded by the presence of the 18C6 molecule in the *t-fau* supercage. This is substantiated by the observation that both samples show a comparable unit cell contraction with pressure. We propose that although the 18C6 molecule has steric bulk, it has inherent molecular flexibility, meaning it can bend in response to the collapse of the supercage. This mirrors the predictions of Fletcher *et al.* [33] from geometric simulations and our previous high-pressure study of the EMT framework containing the 18C6 molecule [34]. However, there is a discrepancy as the flexibility of the filled and empty EMT frameworks is identical, unlike that seen herein for the FAU framework. It is anticipated that the spherical supercage in the FAU framework contracts to a size whereby the 18C6 molecule can no longer flex, and its steric bulk begins to interfere with the mobility of the framework.

The bulk moduli reported herein for zeolite Na-X agree with the anticipated range for open-framework silicates (15–70 GPa) [25] and are comparable to the value reported by Colligan *et al.* [26] for purely siliceous zeolite Y. The similarity in the bulk modulus for zeolites Na-X and Y suggests that the framework Si/Al ratio has little influence on the material's compressibility.

## 3.2. Zeolite RHO

Concerning zeolite RHO, the diffraction patterns for both the empty and filled samples exhibited Bragg peak doubling with the application of pressure. Doubling was observed at 0.2 GPa, with a coexistence of two phases up to 1.1 GPa. This doubling corresponds to the transition from the ambient C-form of *Im-3m* symmetry to the A-form of *I-43m* symmetry, as has been seen previously [15–17]. As mentioned, this transition is characterized by the elliptical distortion of the eight-ring openings in the RHO framework. The presence of the 18C6 molecule does not influence the onset or offset of the symmetry change.

Figure 4*a,b* illustrates the variation in cell parameter *a* with pressure for the empty and filled zeolite RHO accordingly. These figures include the cell parameters for both the C- and A-forms observed, in addition to the edges of their respective flexibility windows simulated using the GASP software. As with zeolite Na-X, both the filled and empty samples of zeolite RHO express a change in gradient at approximately 2.2 GPa, due to the solidification of the pressure-transmitting medium [48].

Concerning the empty zeolite, both the C- and A-forms exist well within the confines of their flexibility windows, demonstrating that there is no distortion of the framework tetrahedra. The placement of the C-form window edges illustrates that the distortion in the A-form permits greater framework mobility that satisfies the rigidity of the tetrahedra. This is due to the reduced symmetry. With decompression, the unit cell expands following the same pathway as the compression, indicating that there is no impedance to the framework mobility from the exerted pressure. Furthermore, the zeolite returns to the ambient C-form with no apparent hysteresis, indicating that the symmetry change is fully reversible.

The reversibility of the symmetry change is also observed for the filled zeolite RHO. Moreover, although the filled sample appears to just leave the flexibility window, the unit cell expansion follows the same pathway as the compression cycle. This indicates that if the framework tetrahedra are distorted at the highest pressure, it is not substantial enough to result in a hysteresis with decompression.

For the ambient C-form, there is an apparent expansion of the flexibility window, indicating that the presence of the 18C6 molecule is improving the flexibility of the framework. This is corroborated by the unusual behaviour observed from the experimentally determined bulk moduli, shown in figure 5. Here, it is shown that the filled C-form of zeolite RHO is significantly more compressible than the empty equivalent. This contrasts typical behaviour, whereby the presence of extra-framework content reduces the zeolite material's softness [25]. Herein, it is understood that due to geometric constraints the framework tetrahedra are effectively moving around the 18C6 molecule with enhanced mobility. This observation of enhanced compressibility with the addition of extra-framework content in zeolites has not been reported previously.

Occupation of the α-cage by the 18C6 is seen to contract the higher density edge of the flexibility window for the lower symmetry A-form. This is to be expected, as it is the steric bulk of the 18C6 molecule which is restricting the extent to which the framework tetrahedra can collapse into the α-cage. Figure 5 displays the unit cell contraction with pressure, illustrating a good agreement between the empty and filled analogues of the A-form. This is substantiated by the congruence in the bulk moduli of these two samples. Furthermore, it demonstrates that with the decrease in the cubic symmetry, the 18C6 molecule is capable of flexing to accommodate the contraction of the framework. Such behaviour mirrors that observed for zeolite Na-X.

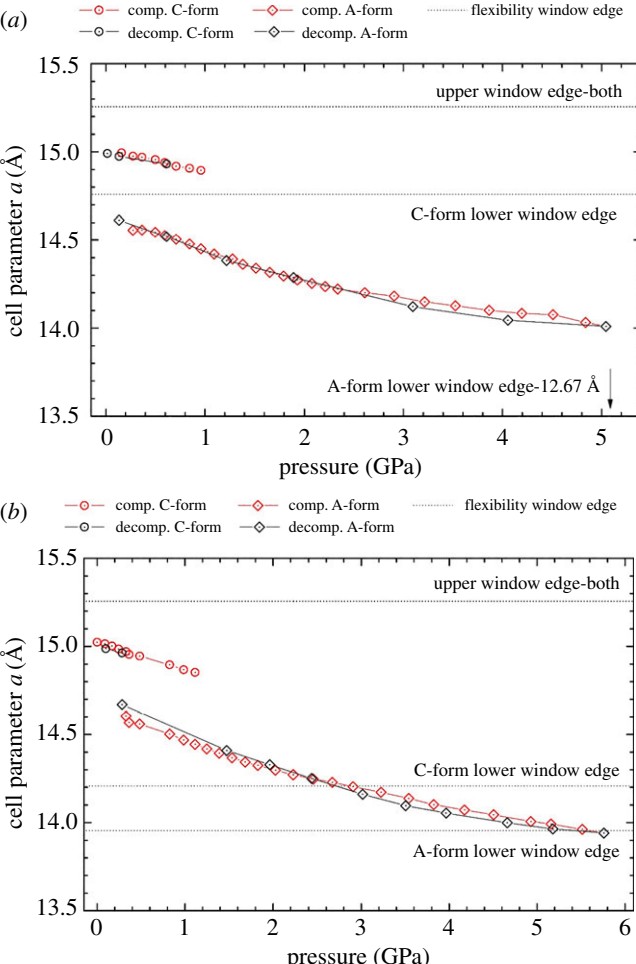

**Figure 4.** Cell parameter $a$ as a function of pressure for ($a$) the empty and ($b$) the filled zeolite RHO. Red data points correspond to the compression cycle, and black to the decompression. Circles correspond to the ambient C-form, and diamonds to the A-form. Also shown are the edges of the flexibility window as simulated in the GASP software.

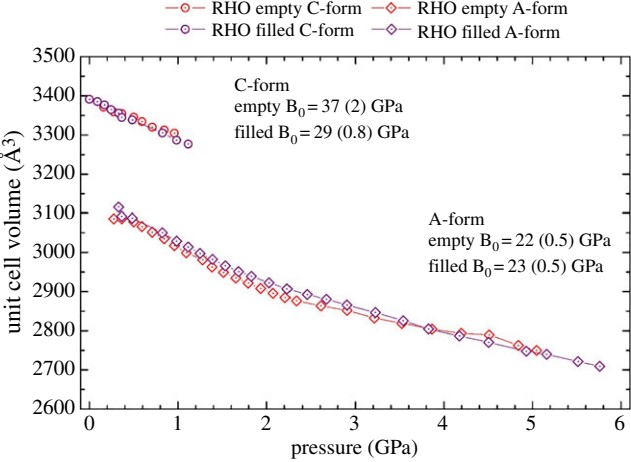

**Figure 5.** The variation of cell volume with increasing pressure for zeolite RHO. The empty analogue is shown in red, and the filled in purple. Circle data points correspond to the C-form, and diamonds to the A-form. Also included are the bulk moduli ($B_0$) of the two forms of the zeolites, using the Birch–Murnaghan second-order fit calculated using the PASCal webtool, as shown in the electronic supplementary material.

## 3.3. Zeolite ZK-5

Compression of zeolite ZK-5 yielded splitting in most of the Bragg peaks from 1.5 GPa onwards for both the empty and filled samples. Bragg peaks where $h$, $k$ and $l$ values of the Miller index are equivalent did

zeolite ZK-5 – 0.0 GPa
cubic phase ($Im\overline{3}m$)

zeolite ZK-5 – 3.0 GPa
tetragonal phase ($I4/mmm$)

**Figure 6.** Unit cell structures of empty zeolite ZK-5 at pressures of 0.0 (cubic) and 3.0 GPa (tetragonal). Simulated in the GASP software using the cell parameters determined from the diffraction data. Oxygen atoms are shown as red spheres, and $SiO_4$ tetrahedra as translucent solids.

not express splitting. Such behaviour was attributed to a phase transition to a tetragonal symmetry. This phase was indexed and refined to the *I4/mmm* space group and characterized by a contraction of cell parameter *c* relative to cell parameters *a* and *b*. The pressure at which the symmetry change occurs was not observed to be influenced by the occlusion of the 18C6 molecule within the framework. Figure 6 shows the GASP simulated structures of the empty zeolite ZK-5 at 0.0 and 3.0 GPa in the cubic and tetragonal symmetries, respectively. This highlights how the tetragonal phase is characterized by a contraction along the *c* axis, in addition to the circular eight-ring openings becoming elliptical. This increasing ellipticity of the eight-ring openings is also observed for zeolite RHO when transitioning from the C- to the A-form as discussed earlier.

Figure 7*a,b* displays the variation in unit cell dimensions for the empty and filled zeolite ZK-5 samples observed throughout the compression. These figures include the unit cell dimensions for both the cubic and tetragonal phases of the sample, alongside their respective flexibility windows simulated using the GASP software. For both samples, the cubic and tetragonal phases were observed within the confines of their simulated windows, indicating no pressure-induced distortions of the framework tetrahedra.

With regards to the windows, the tetragonal window displays some narrowing with the occupation of the α-cage in the KFI framework by the 18C6 molecule. This is to be expected and has been expressed herein by zeolite Na-X and the A-form of zeolite RHO. The reasoning being the same, whereby the steric bulk of the 18C6 molecule is preventing the framework tetrahedra from collapsing into the cavity. Alternatively, the size of the cubic flexibility window is identical between the filled and empty KFI frameworks. This behaviour mirrors what has been observed previously with the EMT framework [33,34] and is similar to the C-form of zeolite RHO herein.

Variation of the unit cell parameters as a function of pressure for the two samples is illustrated in figure 8*a,b*. For the empty zeolite ZK-5, there is a consistent decline in cell parameter *a* with increasing pressure after the transition into the tetragonal phase. However, in the tetragonal phase, there is a sudden contraction in cell parameter *c*, which displays a greater rate of contraction in comparison to the *a* parameter. This demonstrates anisotropic contraction in the tetragonal phase. Furthermore, this suggests that the Daphne oil is no longer transmitting the applied pressure to the zeolite sample hydrostatically.

Similar compression behaviour is observed for the filled zeolite ZK-5 sample; however, the initial decline in the *c* parameter with appearance of the tetragonal phase is less prominent. This is expected to be due to the presence of the 18C6 molecule in the α-cage, which prevents such a collapse along the *c* axis. In addition, there is a marginal expansion of cell parameter *a* with the phase transition. For both the empty and filled samples, cell parameter *c* is more compressible than cell parameter *a*. This is anticipated to be due to the decline in symmetry and consequential axial strain in the unit cell.

The tetragonal phase change is shown to be reversible by the return to the ambient cubic symmetry with decompression. For both the empty and filled samples, the expansion of the unit cell with decompression mostly follows that of the compression. The exception is with the empty zeolite ZK-5, which shows a hysteresis with the return to the cubic phase. Concerning the filled zeolite ZK-5, due to the lack of datapoints it cannot be explicitly confirmed whether a hysteresis is present or not. However, if a

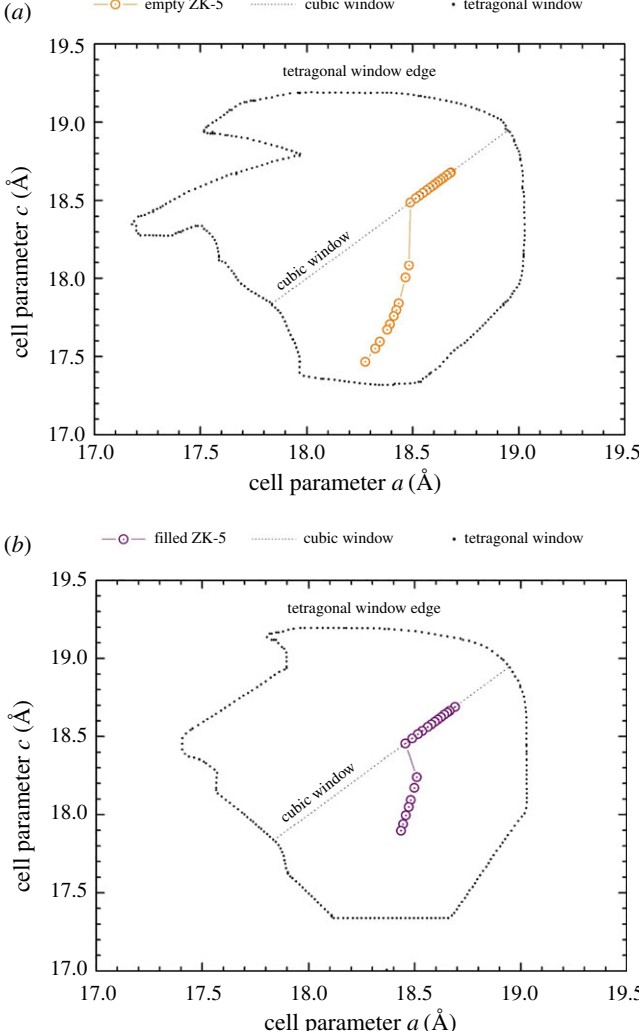

**Figure 7.** Unit cell dimensions of the (*a*) empty and (*b*) filled zeolite ZK-5 throughout compression. Also shown are the extents of the corresponding flexibility windows, as simulated using the GASP software. The black dots display the edge of the tetragonal window, and the fine dashed line the pathway of the cubic window.

hysteresis is present, the returning cubic phase is still observed at a higher pressure compared to that of the empty sample. We postulate that the steric bulk of the occluded 18C6 molecule aids in the reassertion of the cubic symmetry with declining pressure. This is analogous to the structural memory of zeolites exerted by hydrated cations in the FAU framework [49,50], as mentioned previously.

Figure 9 displays the unit cell volume contraction as a function of pressure for both samples of zeolite ZK-5, as well as the bulk moduli of the corresponding phases. Owing to experimental time constraints, the filled zeolite ZK-5 was depressurized before the onset of pressure-induced amorphization. In the cubic phase, the presence of the 18C6 molecule in the α-cage appears to make the zeolite more compressible. Such behaviour is comparable to that seen for the C-form of zeolite RHO. Consequently, the same rationale can be used, whereby due to the geometric constraints, the framework tetrahedra express improved mobility. This further reinforces that there is a geometric congruence between the 18C6 molecule and the α-cage. Moreover, both samples of the cubic phase show a negative pressure derivative, indicative of pressure-induced softening [47]. Previously, Fang & Dove [51] have simulated and predicted such dynamic instabilities for zeolite ZK-5 and other cubic zeolites. However, the pressure derivatives observed herein for zeolite ZK-5 are significantly larger than anticipated.

With the transition to the tetragonal phase, both the empty and filled zeolite ZK-5 samples show increased compressibility. As with zeolite RHO, this behaviour demonstrates the enhanced compressibility is facilitated by the transition to a lower symmetry. Within the tetragonal phase, the filled zeolite ZK-5 is shown to be less compressible, as is expected. In this case, the steric bulk of the 18C6 molecule is preventing the collapse of the framework. This is corroborated by the data itself, which shows how the filled zeolite ZK-5 displays a significantly more expanded unit cell volume than the empty analogue, at the same pressure.

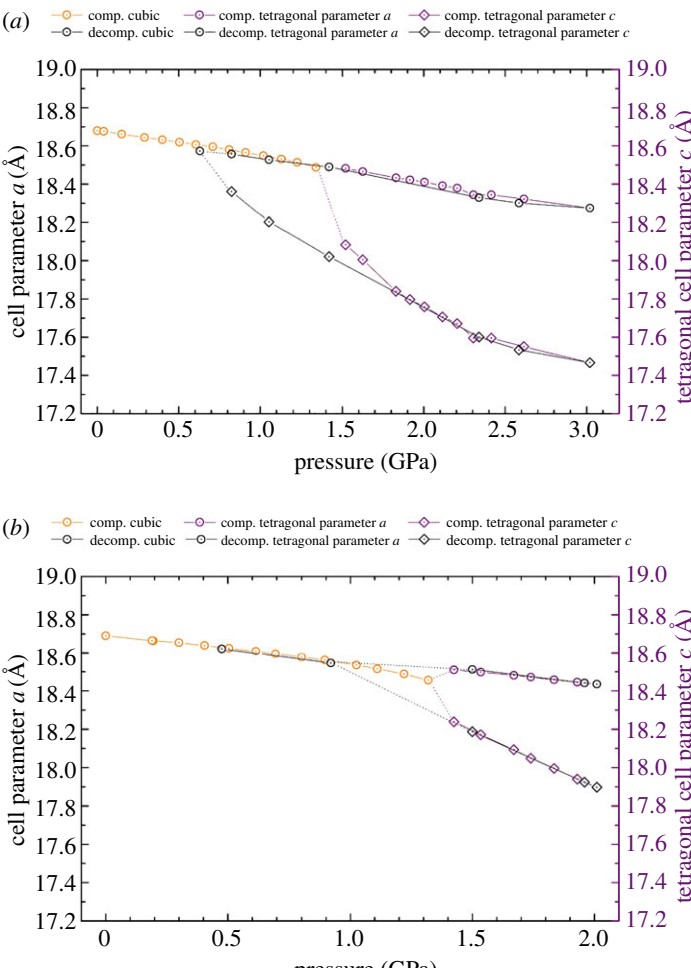

**Figure 8.** Cell parameter *a* and tetragonal cell parameter *c* as a function of pressure for (*a*) the empty and (*b*) the filled zeolite ZK-5. Circles correspond to the *a* parameter, and diamonds to the tetragonal *c* parameter. Data points shown in orange refer to the cubic phase, purple the tetragonal phase and black to the decompression cycle of both phases. The dotted lines indicate the cell parameter shift between the two phases.

# 4. Discussion

From the results herein, it is apparent that zeolites RHO and ZK-5 are comparable with regards to their compression behaviour. As these two zeolites share several symmetry and topological features, this highlights how the compression mechanism is dominated by framework geometry. Specifically, for zeolites RHO and ZK-5, there is the observation that in their ambient symmetries, the occlusion of the 18C6 molecule appears to enhance the mechanical softness of the framework. An observation that is further supported by the flexibility windows, which expand or are unchanged with occlusion of 18C6. This contrasts convention that extra-framework content should reduce the compressibility [11,25–28].

This behaviour has been rationalized as an influence of geometric constraints imposed by the 18C6 molecule. Such constraints being unique to the α-cage, suggesting an exclusive relationship between the 18C6 molecule and the α-cage geometry. Consequently, the 18C6 molecule expresses behaviour indicative of a geometric template in the synthesis of zeolites RHO and ZK-5. By contrast, zeolite Na-X shows a contraction of the flexibility window with 18C6 occlusion, but no influence on the mechanical softness due to the 18C6 molecule's own inherent flexibility. This is believed to be due to the fact that zeolite Na-X has a larger aperture (12-ring) and cavity size compared to zeolites RHO and ZK-5, meaning there is more available space for the 18C6 molecule. Furthermore, this suggests that in the assembly of zeolite Na-X, the 18C6 expresses weak interactions with the framework, indicative of a space-filling species.

In addition, zeolites RHO and ZK-5 also show a reduction in symmetry with pressure and increase in mechanical softness, which zeolite Na-X does not. Although topologically zeolites RHO and ZK-5 share an α-cage, the reason for this distinction is clearer upon comparison to other cubic and tetragonal zeolites.

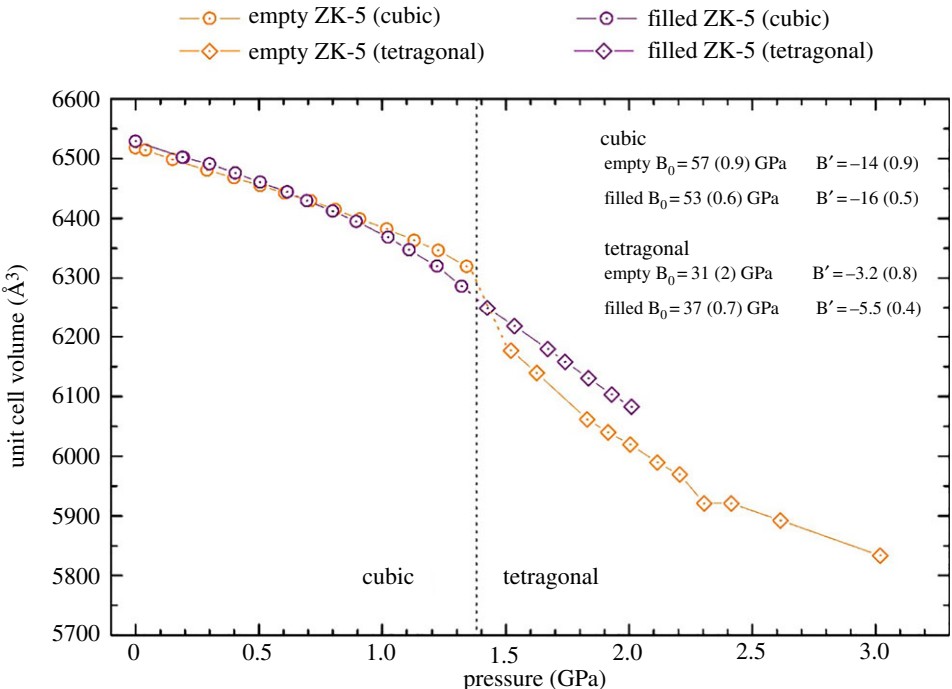

**Figure 9.** The variation of cell volume with increasing pressure for zeolite ZK-5. The empty analogue is shown in orange, and the filled in purple. Circle data points correspond to the cubic phase, and diamonds to the tetragonal phase. The approximate position of the phase transition is shown with the dotted line. Also included are the bulk moduli ($B_0$) and pressure derivatives ($B'$) of the two phases of the zeolites, using the Birch–Murnaghan third-order fit calculated using the PASCal webtool, as shown in the electronic supplementary material.

**Table 2.** A summary of cubic and tetragonal zeolites that have been studied with high-pressure X-ray diffraction. Contained are the space groups at ambient conditions and observed under compression.

| zeolite | framework type | ambient space group | pressure-induced space group | |
|---|---|---|---|---|
| analcime | ANA | *Ia-3d* | *P-1* | [20] |
| leuctie | ANA | *I4₁/a* | *P-1* | [21] |
| pollucite | ANA | *Ia-3d* | *P-1* | [22] |
| wairakite | ANA | *I2/a* | *P-1* | [23] |
| zeolite ZK-5 | KFI | *Im-3m* | *I4/mmm* | |
| zeolite RHO | RHO | *Im-3m* | *I-43m* | [15–17,52] |
| sodalite | SOD | *Im-3m* | *I-43m* | [4,53,54] |
| zeolite A | LTA | *Fm-3c* | — | [28,55,56] |
| zeolite Na-X | FAU | *Fd-3m* | — | [57] |
| zeolite Y | FAU | *Fd-3m* | — | [26] |

Table 2 summarizes the findings herein alongside the results of other zeolites analysed using high-pressure X-ray diffraction in the literature. These results highlight that a phase change with pressure is only observed for zeolites with body-centred symmetry. This is even true when the ambient space group differs between zeolites of the same framework type, as is the case for the ANA-type zeolites [20–23]. For ANA-type zeolites, the lower symmetry form is consistently triclinic, and according to geometric simulations by Wells *et al.* [24], it is the edge of the flexibility window which controls these pressure-induced symmetry changes. However, herein for the higher symmetry/low-pressure forms of zeolites RHO and ZK-5, we see the pressure-induced symmetry change before the window edge is reached.

We propose that such pressure-induced symmetry changes are intrinsic to body-centred cubic zeolites. In the ambient form, the zeolite symmetry is so high that the framework is restricted in the degrees of freedom in

which the tetrahedra can tilt while maintaining the symmetry. It is geometrically favourable for the symmetry to be lowered, permitting continued compression without compromising the rigid shape of the framework tetrahedra. Consequently, we predict that other body-centred cubic aluminosilicate zeolites will also express pressure-induced symmetry changes upon compression. Paulingite being a candidate zeolite. Furthermore, we also anticipate that other non-body-centred aluminosilicate zeolites, such as tschörtnerite and Linde Type N, will show no such change in symmetry.

## 5. Conclusion

High-pressure powder X-ray diffraction data demonstrates a pressure-induced symmetry change from cubic to tetragonal for zeolite ZK-5. For zeolites RHO and ZK-5, the presence of 18C6 within the framework cavities does not influence the pressure at which symmetry transitions occur. In both cases, the 18C6 containing zeolite is more compressible than the empty framework, suggesting geometric match-up with the α-cage. For zeolite Na-X, the 18C6 molecule expresses negligible influence on the mechanical softness of the framework. From observing the influence of the 18C6 molecule on framework dynamics, it is discerned that 18C6 behaves as a space-filling species in the preparation of zeolite Na-X and employs geometric structure direction for zeolites RHO and ZK-5.

In conjunction with previous high-pressure analysis in the literature, we propose that pressure-induced reductions in symmetry are intrinsic to body-centred cubic zeolites. The phase transition is geometrically driven and allows the framework to be compressed further without distorting the framework tetrahedra, which is energetically costly.

Data accessibility. All data created during this study are available free of charge from the University of Bath data archive at https://doi.org/10.15125/BATH-00580 [58].

Authors' contributions. The study was conceived by A.S. and A.N. The data were collected by A.N., M-L.U.C., Z.L.J. and A.S. with the assistance of I.E.C. on the ID15B beamline at the ESRF. The geometric simulations in GASP were performed by M-L.U.C. Pawley refinements using the TOPAS Academic software were undertaken by A.N. All authors, including S.A.W. and P.R.R., were involved in the data interpretation and editing of the manuscript. The original manuscript was written by A.N., with input and approval from all authors. All authors gave final approval for publication.

Competing interests. We declare we have no competing interests.

Funding. This study was funded by the Royal Society under the 'New flexible frameworks for catalysis, energy materials and nanotechnology' URF grant awarded to A.S. Further funding was contributed by the EPSRC under the EP/K004956/1 grant, of whom P.R.R. is the principal investigator. S.A.W. acknowledges funding from the ERC under the European Union's Horizon 2020 Research and Innovation Programme (grant agreement no. 648283 'GROWMOF'). M-L.U.C. would like to thank the National Research Foundation of South Africa for funding the research project and the Commonwealth Scholarship Commission in the United Kingdom for funding the research visit to the University of Bath, United Kingdom.

Acknowledgements. The high-pressure X-ray diffraction data herein was collected on beamline ID15B at the European Synchrotron Radiation Facility (ESRF), Grenoble, France. We thank the ESRF Council for accepting our research proposal.

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
