## [Reviewer comments · Royal Society Open Science]

Review History

RSOS-182158.R0 (Original submission)

Review form: Reviewer 1

Is the manuscript scientifically sound in its present form?

Yes

Are the interpretations and conclusions justified by the results?

Yes

Is the language acceptable?

Yes

Is it clear how to access all supporting data?

Yes

Do you have any ethical concerns with this paper?

No

Have you any concerns about statistical analyses in this paper?

No

Recommendation?

Accept as is

Comments to the Author(s)

This paper addresses a sparsely investigated, yet very interesting, area of research pertaining to the influence of pressure on the symmetry of zeolites prepared by using 18-crown-6 ether as an organic additive. Three zeolites, that is Na-X, RHO and ZK-5 have been studied, along with their empty analogs (zeolites for which the crown ether has been removed by thermal treatment). A systematic study of their crystal structure performed by high-pressure X-ray diffraction revealed that the synthesis of zeolites has a significant impact on their crystal structure and, consequently, on the flexibility of their networks. In particular, the authors suggest that the crown ether molecule acts as a geometric template in the synthesis of RHO and ZK-5 zeolites, behavior which is not displayed in the synthesis of the Na-X zeolite. Such a behavior has been tentatively ascribed to the body-centered structure of the zeolites whereby compression of the structural framework is allowed without influencing the rigidity of the networks. Based upon the experimental results, the authors postulate that such a behavior will be absent in the case of non-body-centered aluminosilicate zeolites. The manuscript is well-articulated, the discussion and conclusions being well supported by the experimental data. Perhaps it will be useful for the reader to know what is the experimental error in the determination of the unit cell parameters and cell volumes with the pressure. based upon the foregoing, I recommend the publication of this manuscript after the authors address the comments indicated above.

Review form: Reviewer 2

Is the manuscript scientifically sound in its present form?

Yes

Are the interpretations and conclusions justified by the results?

Yes

Is the language acceptable?

Yes

Is it clear how to access all supporting data?

Yes

Do you have any ethical concerns with this paper?

No

Have you any concerns about statistical analyses in this paper?

No

Recommendation?

Accept with minor revision (please list in comments)

Comments to the Author(s)

Dear Authors,

This a great work in zeolites world. I have a couple thoughts to share.

First, do you think the discrepancy of compressibility between ZK-5, RHO, and NaX is simply due to the larger beta-cage, that is 12-ring vs. 8-ring? Also, it's well accepted that the zeolite pores do possess certain flexibility, it would be nice to reference some simulation or other experimental data for comparison.

Thanks.

Decision letter (RSOS-182158.R0)

22-May-2019

Dear Dr Sartbaeva:

Title: Pressure-induced symmetry changes in body-centred cubic zeolites
Manuscript ID: RSOS-182158

Thank you for submitting the above manuscript to Royal Society Open Science. On behalf of the Editors and the Royal Society of Chemistry, I am pleased to inform you that your manuscript will be accepted for publication in Royal Society Open Science subject to minor revision in accordance with the referee suggestions. Please find the reviewers' comments at the end of this email.

The reviewers and handling editors have recommended publication, but also suggest some minor revisions to your manuscript. Therefore, I invite you to respond to the comments and revise your manuscript.

Please also include the following statements alongside the other end statements. As we cannot publish your manuscript without these end statements included, if you feel that a given heading is not relevant to your paper, please nevertheless include the heading and explicitly state that it is not relevant to your work. We have included a screenshot example of the end statements for reference.

- Acknowledgements

- Funding statement

Please include a funding section after your main text which lists the source of funding for each author.

Because the schedule for publication is very tight, it is a condition of publication that you submit the revised version of your manuscript before 31-May-2019. Please note that the revision deadline will expire at 00.00am on this date. If you do not think you will be able to meet this date please let me know immediately.

To revise your manuscript, log into <https://mc.manuscriptcentral.com/rsos> and enter your Author Centre, where you will find your manuscript title listed under "Manuscripts with

Decisions". Under "Actions," click on "Create a Revision." You will be unable to make your revisions on the originally submitted version of the manuscript. Instead, revise your manuscript and upload a new version through your Author Centre.

Best wishes,
Dr Laura Smith
Publishing Editor, Journals

RSC Associate Editor:
Comments to the Author:
(There are no comments.)

RSC Subject Editor:
Comments to the Author:
(There are no comments.)

Reviewer comments to Author:
Reviewer: 1

Comments to the Author(s)

This paper addresses a sparsely investigated, yet very interesting, area of research pertaining to the influence of pressure on the symmetry of zeolites prepared by using 18-crown-6 ether as an organic additive. Three zeolites, that is Na-X, RHO and ZK-5 have been studied, along with their empty analogs (zeolites for which the crown ether has been removed by thermal treatment). A systematic study of their crystal structure performed by high-pressure X-ray diffraction revealed that the synthesis of zeolites has a significant impact on their crystal structure and, consequently, on the flexibility of their networks. In particular, the authors suggest that the crown ether molecule acts as a geometric template in the synthesis of RHO and ZK-5 zeolites, behavior which is not displayed in the synthesis of the Na-X zeolite. Such a behavior has been tentatively ascribed to the body-centered structure of the zeolites whereby compression of the structural framework is allowed without influencing the rigidity of the networks. Based upon the experimental results, the authors postulate that such a behavior will be absent in the case of non-body-centered aluminosilicate zeolites. The manuscript is well-articulated, the discussion and conclusions being well supported by the experimental data. Perhaps it will be useful for the reader to know what is the experimental error in the determination of the unit cell parameters and cell volumes with the pressure. based upon the foregoing, I recommend the publication of this manuscript after the authors address the comments indicated above.

Reviewer: 2

Comments to the Author(s)
Dear Authors,

This a great work in zeolites world. I have a couple thoughts to share.
First, do you think the discrepancy of compressibility between ZK-5, RHO, and NaX is simply due to the larger beta-cage, that is 12-ring vs. 8-ring? Also, it's well accepted that the zeolite pores do possess certain flexibility, it would be nice to reference some simulation or other experimental data for comparison.

Thanks.

Author's Response to Decision Letter for (RSOS-182158.R0)

See Appendix A.

RSOS-182158.R1 (Revision)

Review form: Reviewer 2

Is the manuscript scientifically sound in its present form?

Yes

Are the interpretations and conclusions justified by the results?

Yes

Is the language acceptable?

Yes

Do you have any ethical concerns with this paper?

No

Recommendation?

Accept as is

Comments to the Author(s)

The revision appears good to me.

Decision letter (RSOS-182158.R1)

24-Jun-2019

Dear Dr Sartbaeva:

Title: Pressure-induced symmetry changes in body-centred cubic zeolites

Manuscript ID: RSOS-182158.R1

It is a pleasure to accept your manuscript in its current form for publication in Royal Society Open Science. The chemistry content of Royal Society Open Science is published in collaboration with the Royal Society of Chemistry.

RSC Associate Editor:
Comments to the Author:
(There are no comments.)

RSC Subject Editor:
Comments to the Author:
(There are no comments.)

Reviewer(s)' Comments to Author:
Reviewer: 2

Comments to the Author(s)
The revision appears good to me.

Appendix A

Reviewer: 1

Comments to the Author(s)

This paper addresses a sparsely investigated, yet very interesting, area of research pertaining to the influence of pressure on the symmetry of zeolites prepared by using 18-crown-6 ether as an organic additive. Three zeolites, that is Na-X, RHO and ZK-5 have been studied, along with their empty analogs (zeolites for which the crown ether has been removed by thermal treatment). A systematic study of their crystal structure performed by high-pressure X-ray diffraction revealed that the synthesis of zeolites has a significant impact on their crystal structure and, consequently, on the flexibility of their networks. In particular, the authors suggest that the crown ether molecule acts as a geometric template in the synthesis of RHO and ZK-5 zeolites, behavior which is not displayed in the synthesis of the Na-X zeolite. Such a behavior has been tentatively ascribed to the body-centered structure of the zeolites whereby compression of the structural framework is allowed without influencing the rigidity of the networks. Based upon the experimental results, the authors postulate that such a behavior will be absent in the case of non-body-centered aluminosilicate zeolites. The manuscript is well-articulated, the discussion and conclusions being well supported by the experimental data. Perhaps it will be useful for the reader to know what is the experimental error in the determination of the unit cell parameters and cell volumes with the pressure. based upon the foregoing, I recommend the publication of this manuscript after the authors address the comments indicated above.

We thank the reviewer for their positive comments and their recommendation. To address their comment regarding the errors in unit cell parameters and cell volumes we have edited the manuscript as follows:

In the Materials and Methods section we have included the following sentences:

"The error in the calculated unit cell parameters were determined in the TOPAS Academic software. It was seen that the errors were consistently $<0.004 \text{ \AA}$ and had negligible influence on the trends observed. Tables containing the full list of calculated errors for the unit cell parameters and cell volumes are contained in the Supplementary Information (SI)."

As mentioned, we have thus updated the supplementary information (SI) to include tables of the error of the unit cell parameters and volumes calculated. For completion, we have also made mention of the experimental error in the pressure determination, and included these errors in the SI as well. The added sentence in the Materials and Methods section is as follows: "The experimental error in pressure was estimated at 0.1 GPa in the 0.0-3.0 GPa region of the compression cycle. Above 3.0 GPa, and throughout the entirety of the decompression cycle the error was estimated at 0.4 GPa."

Reviewer: 2

Comments to the Author(s)

Dear Authors,

This a great work in zeolites world. I have a couple thoughts to share.

First, do you think the discrepancy of compressibility between ZK-5, RHO, and NaX is simply due to the larger beta-cage, that is 12-ring vs. 8-ring?

We thank the reviewer for their compliments and the issues they raise. The difference in compressibility is due to the different aperture and cavity sizes present in zeolites Na-X, ZK-5 and RHO. Due the larger cavity in zeolite Na-X there is more space available for the 18C6. In order to clarify this point that we were trying to make we have included the following sentence to the Discussion section:

"This is believed to be due to the fact that zeolite Na-X has a larger aperture (12-ring) and cavity size compared to zeolites RHO and ZK-5, meaning there is more available space for the 18C6 molecule. Furthermore, this suggests that in the assembly of zeolite Na-X the 18C6 express weak interactions with the framework, indicative of a space-filling species."

Also, it's well accepted that the zeolite pores do possess certain flexibility, it would be nice to reference some simulation or other experimental data for comparison.

We thank the reviewer for noting this. Indeed, we have included multiple references regarding zeolite framework flexibility, and a mixture of simulated and experimental work. For example: We have compared how extra-framework content influences compression [11, 13, 25-28, 49, 50], how the data compares to what has been seen previously for the same zeolites [15-17, 26, 52], how the flexibility windows of the zeolites compare to another zeolite (EMC-2) made with 18-crown-6 ether [33]. In Table 2, we have also made numerous citations to experimental and simulated work on cubic zeolites.

To improve the amount of referencing and comparison in the manuscript we have made the following changes:

- 1) In the Zeolite Na-X section, we have added the following sentence: "The linear unit cell compression is comparable to the results seen by Colligan et al. [26] for other FAU-type zeolites of varying Si/Al ratio in different pressure-transmitting media."*
- 2) We have included an additional reference ([34] in the manuscript), referring to a previous high pressure study on zeolite EMC-2.. This reference has been added to the Introduction, Zeolite Na-X and Zeolite ZK-5 sections, in order to further compare the flexibility and compression behaviour with the occlusion of 18C6. In the zeolite Na-X section a sentence has been adjusted to state as follows: "This mirrors the predictions of*

Fletcher et al. [33] from geometric simulations and our previous high pressure study of the EMT framework containing the 18C6 molecule [34]."

- 3) *In the Discussions section we have included some further discussion comparing the flexibility and phase transitions seen in ANA-type zeolites and zeolites RHO and ZK-5. The nature of all the symmetry transitions are summarised in table 2. The included discussion is as follows: "This is even true when the ambient space group differs between zeolites of the same framework type, as is the case for the ANA-type zeolites [20-23]. For ANA-type zeolites the lower symmetry form is consistently triclinic, and according to geometric simulations by Wells et al. [24] it is the edge of the flexibility window which controls these pressure-induced symmetry changes. However, herein for the higher symmetry/low pressure forms of zeolites RHO and ZK-5 we see the pressure-induced symmetry change before the window edge is reached."*